# Frequency of physical activity during leisure time and variables related to pain and pain medication use in Spanish adults: A cross-sectional study

Ángel Denche-Zamorano[1], Raquel Pastor-Cisneros[1], Antonio Castillo-Paredes[2]*, José Carmelo Adsuar-Sala[1], Diana Salas-Gómez[3]

1 Promoting a Healthy Society (PHeSO) Research Group, Faculty of Sport Sciences, University of Extremadura, Cáceres, Spain, 2 Grupo AFySE, Investigación en Actividad Física y Salud Escolar, Escuela de Pedagogía en Educación Física, Facultad de Educación, Universidad de Las Américas, Santiago, Chile, 3 Escuelas Universitarias Gimbernat (EUG), Physiotherapy School Cantabria, University of Cantabria, Torrelavega, Spain

* acastillop85@gmail.com

**Data Availability Statement:** Microdata from the European Health Survey in Spain are available on the website of the Spanish National Statistics

## Abstract

### Background

Physical inactivity has been identified as a risk factor for pain.

### Objective

The main objective was to analyze the relationships between leisure time physical activity frequency (PAF) and pain prevalence, pain level, pain impairment, daily life pain impairment, and analgesic use in Spanish adults. In addition, risk factors such as sex, body mass index, marital status and social class were assessed for these pain variables in addition to PAF.

### Method

Cross-sectional study based on data from the 2014 and 2020 European Health Surveys in Spain residents. The Chi-square test was used to analyze the relationship of dependence between the variables of interest. A correlation study calculating Spearman's rho and a multiple logistic regression were performed to assess risk factors for pain variables.

### Results

20,113 and 19,196 subjects with a median age of 49 and 52 years old in 2014 and 2020 European Health Surveys, respectively, were analyzed. Dependence relationships were found between PAF and pain variables (p<0.001). The prevalence of: pain, high levels of pain, pain impairment, high level of pain impairment and use analgesic were higher in the inactive population than in the rest of the PAF levels (36.7–53%) vs (18.6–44.3%), p<0.05. Weak correlations were found between PAF and pain variables (-0.177 ≤ Rho ≤ -0.107) (p<0.001). Logistics regression show that being active or very active reduces the odds of

Institute. INEbase / Society /Health /European Survey of Health in Spain / Results 2014: https://www.ine.es/ftp/microdatos/encuersalud/datos_2014_adultos.zip. 2020: https://www.ine.es/ftp/microdatos/encuersalud/datos_2020_individual.zip.

**Funding:** The APC was funded by the Open Access Program of Universidad de Las Américas. The author A.D.-Z. (FPU20/04201) was supported by a grant from the Spanish Ministry of Education, Culture and Sport. Grants FPU20/04201 funded by MCIN/AEI/10.13039/501100011033 and, as appropriate, by "European Social Found Investing in your future" or by "European Union NextGenerationEU/PRTR". The author R.P.-C. was supported with a grant by the Valhondo Calaff Foundation (Caceres, Spain). The funders had no role in study design, data collection and analysis, decision to publish, or preparation of the manuscript."

**Competing interests:** The authors declare no conflict of interest.

pain, the intensity of pain and being affected in daily activities due to pain by 0.524 to 0.776 times. Likewise, being active or very active reduces the odds of taking pain medication by 0.661 to 0.755 times. Also age, low social class, being a woman, and being obese increase of odds of pain, pain affectation and use of analgesics in both surveys by 1.008 to 2.202 times.

## Conclusions

Physical inactivity was related to a higher prevalence of: pain, higher levels of pain, higher pain involvement and higher analgesic use. In addition, lower social class, being female, older age, and obesity were factors for higher odds of pain, pain involvement, and analgesic use in both surveys.

## Background

The World Health Organization (WHO) recognizes that chronic pain is a public health problem of global magnitude [1], with data indicating that one-third of the world's population suffers from some form of pain [2]. Currently, the International Association for the Study of Pain (IASP) defines pain as an unpleasant sensory and emotional experience associated with or similar to that associated with actual or potential tissue damage [3]. Chronic pain is defined as pain present most of the time for a period of 3 months [4]. Pain is the main reason people seek medical care, mainly due to osteoarthritis, back pain, and headaches [5]. Prevalence rates of pain are higher in women, people of low socioeconomic status, and people residing in rural settings [6]. Among the main effects of pain is the worsening of personal relationships and self-esteem, being associated with a higher risk of suicide, and substance use [7] as well as being related to a lower life expectancy [8].

At present, there are various recommendations for meeting the minimum physical activity requirements. On the one hand, it is indicated that a minimum of 30 minutes of moderate-intensity physical activity, five days per week, or 20 minutes of more intense physical activity, three days per week, is necessary [9]. On the other hand, WHO recommends that adults (18–64 years) should perform at least 150 minutes of moderate-intensity aerobic physical activity throughout the week, 75 minutes of vigorous-intensity aerobic physical activity, or a combination of both [10]. However, available data suggest that 27.5% of the world's population does not meet either of these minimum recommendations [11], and in particular, one in three adults in the European Union are physically inactive [12].

Physical inactivity has been identified as a risk factor for developing chronic pain [13]. Chronic pain has high treatment costs, mainly impacting daily activities and quality of life [14]. However, it has been shown that physical activity and exercise have a positive relationship with the reduction of the intensity of chronic pain, leading to an improvement in quality of life and physical function [15], with the main premise that it is controlled and/or supervised by a professional [16]. It has been shown that depending on the type of exercise performed, a greater contribution is made to alleviating localized pain [17, 18]. For example, aquatic exercise is associated with an improvement in chronic back pain [17], while combat sports such as taichi specifically can reduce the pain of arthritis [18] for aerobic and strength training had beneficial effects on pain reduction in patients with fibromyalgia [19].

Regarding the relationship between PA and pain, a previously study conduct in 17,777 Spanish residents [20] found that inactive people also had the highest prevalence of pain, were more affected by pain in their daily lives, and took more medication [20]. In addition, a recent study also observed in a sample of Spanish residents that doing physical activity during leisure

time decreased the probability of suffering from chronic low back pain. However, performing physically demanding occupational tasks increased the probability of suffering from chronic low back pain [21].

However, none of the cited scientific evidence explores the association between physical activity frequency during leisure time and pain. Additionally, there is a lack of research on how this variable may affect chronic pain, including factors such as level of impairment or medication use. Considering previous studies that demonstrate the significance of physical activity frequency during leisure time, it is necessary to investigate its potential correlation with pain and related variables in the Spanish population. This investigation may aid in the prevention and treatment of pain through public health measures.

Therefore, the main objective was to analyze the relationships between leisure time physical activity frequency (PAF) and pain prevalence, pain level, pain impairment, daily life pain impairment, and analgesic use in Spanish adults. In addition, risk factors such as sex, body mass index, marital status and social class were assessed for these pain variables in addition to PAF. The starting hypothesis of the present study was: PAF in leisure time is related to pain in the Spanish population.

## Materials and methods

### Study design and setting

This was a cross-sectional study carried out using data from the European Health Interview Survey (EHIS) in Spain in 2014 (EHISS 2014) [22] and 2020 (EHISS 2020) [23]. The aim of EHIS is measure the health status, health determinants (lifestyle-related, such as sedentary lifestyles) and the use of health care, services and limitations in access to them of European Union citizens, in a harmonized way and with a high degree of comparability between Member States (MS). Coordinated by the European Statistical Office (Eurostat), it is carried out by those responsible for each MS, in the case of Spain, by the National Statistics Institute (INE) in collaboration with the Spanish Ministry of Health (MSS). The target population was those over 15 years of age, residing in non-institutionalized households in the MS. The questionnaire consists of four modules: health status, health determinants, health care, and background variables (sociodemographic). In Spain, interviewers trained and accredited by the INE; The EHISS 2014 (between January 2014 and January 2015) [22] and the EHISS 2020 (between July 2019 and July 2020) [23] administered the questionnaire by a personal interview.

### Sample size

Previously, participants were selected using a three-stage stratified random sampling system (First stage units are census tracts. The second stage units are the main family households. In the third stage, one adult (15 years old or over) is selected from each household to fill in the Adult Questionnaire); they were informed of their selection, the nature of the survey, the anonymous treatment of the data, and consent was obtained from those who agreed to participate in the survey. The methodologies of both surveys describe all the information on the regulations that govern them, on the calculation of the necessary sample, on the treatment of the data, on the explanation of the variables, on the treatment of missing data, among other aspects.

### Ethics approval and consent to participate

Not applicable. The data were obtained from non-confidential, open-access public records published by the INE, no official ethics committee oversight and authorization was required. However, the 2014 and 2020 EHSS were regulated by Commission Regulation (EU) 141/2013

of 19 February 2013 and Commission Regulation (EU) 255/2018 of 19 February 2019, implementing Framework Regulation 1338/2008.

**Participants.** Data from both the EHISS 2014 and the EHISS 2020 were download from public files provided by the INE in the INE website: EHISS 2014 and EHISS 2020. These included anonymous responses from 22,842 (EHISS 2014) and 22,072 (EHISS 2020), over 15 years of age and resident in Spain. On this sample, our research had the following selection criteria: 1) being older than 18 years; 2) younger than 80 years; 3) and having submitted responses to the item corresponding to PAF. Therefore, the following were excluded from the EHISS: 521 participants under 18 years of age, 2,175 participants of same or over 80 years of age, and 33 participants with no information on PAF in leisure time (item Q.112). Likewise, from the EHISS 2020 were excluded 503 participants younger than 18 years, 2,356 participants of same or older than 80 years, and 17 participants with no response to item Q.112. The final sample was 39,309 participants between 18–79 years (EHISS 2014: 20,113; EHISS 2020: 19,196). Fig 1 shows the flowchart of selection of participants and the exclusions made in the analyzes of each specific item.

**Variables.** From the downloaded files (EHISS 2014 and 2020), data were extracted for the following variables of interest:

*Independent or predictors variables*. *Sociodemographic variables*: *Age* from item AGEa, in years, *Sex* from item SEXa: Men or Women. The body mass index (BMI) Group variable was obtained from item BMIa. This variable grouped participants into Underweight (BMI<18.5), Normal (BMI between 18.5 and ≤25), Overweight (BMI between 25 and ≤30), and Obesity (BMI >30). BMI computed from self-reported weight and height is a valid measure in men and women across different socio-demographic group [24].

For analysis with this variable, 1,434 participants were excluded: 699 in the EHISS 2014, and 735 in the EHISS 2020, due to a lack of data on this item.

*Socioeconomic variables*: Another variable was *Social Class* (based on the item CLASE_PR, which grouped the participants according to the 2011 National Classification of Occupations (CNO 2011) [22, 23, 25] into 6 categories according to the average proportion of workers in the company and the academic degree held: I, II, III, IV, V, and VI). According to this classification, the social classes of people responsible for companies with a higher proportion of workers or people with a higher level of education would correspond to the classes categorized with lower numbers and vice versa. Thus, social class I was made up of those persons who were directors or managers of establishments with 10 or more workers under their charge and conventionally by professionals associated with higher studies, and class VI in this case was made up of unskilled workers. Additional file 1 shows a more comprehensive description of this classification. For the analyses that included this variable, 1,105 participants were excluded: 414 from 2014 EHISS and 691 from 2020 EHISS for not submitting data.

*Civil Status* from item Q.4b, grouped participants according to their civil status: Single, Married, Widowed, Legally separated, Divorced, NS/NC. For analyses that included this variable, 80 participants were excluded: 20 from EHISS 2014 and 60 from EHISS 2020.

*Physical activity variables*: *Physical Activity Frequency* (PAF) in leisure time from item Q.112: Which of these possibilities best describes the frequency with which you do some physical activity in your free time? With the following possible answers: I do not exercise. I spend my free time almost sedentary (In this study it was considered as "Inactive"), I do some occasional physical activity or sport (It was considered as "Occasionally"), I do physical activity several times a month (It was considered as "Active") and I do sport or physical training several times a week (It was considered as "Very active").

*Dependent variables*. *Pain-related variables*: *Pain Level from* item Q.45: During the last 4 weeks, what degree of pain have you experienced? None, Very Mild, Mild, Moderate, Severe,

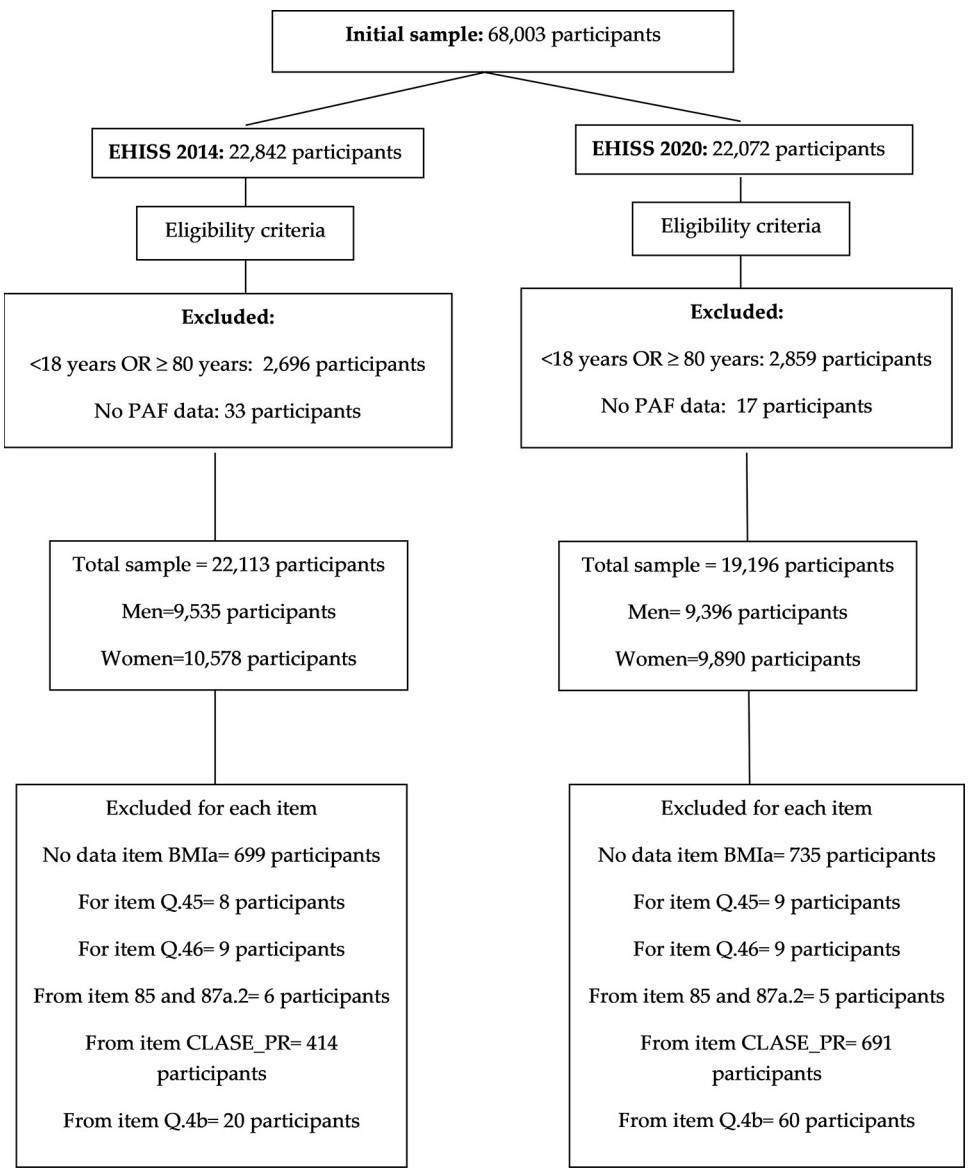

**Fig 1. Flowchart outlining the study sample's selection.**

Extreme, Don't know, or Don't answer (NS/NC). For the analyses that included this variable, 17 participants were excluded: 8 from EHISS 2014 and 9 from EHISS 2020 for answering NS/NC to this item.

*Pain Affect Level* was also extracted (from item Q.46: During the last 4 weeks, to what extent did pain affect your daily activities? Nothing, A little, Moderately, Fairly, A lot, NS/NC). For the analyses that included this variable, 18 participants were excluded: 9 in each of the surveys for answering NS/NC to this item).

*Created variables*: *Pain Medication* is a combination of two items: Q.85: During the last 2 weeks, have you taken any medication that was prescribed to you by a doctor? Yes or No; and from item Q.87a.2: Next, I am going to read you a list of types of medications, please tell me which one(s) you have taken in the last 2 weeks and which one(s) was prescribed to you by a doctor: Pain medication? Yes, No, or NS/NC. For this new variable, "Yes" was considered in

those cases in which the answer to item Q.87a.2 was "Yes". Likewise, "No" was considered in two situations: 1) with a "No" response to item Q.85, or 2) with the participant's response "Yes" to item 85 and "No" to item Q.87a.2). In analyses that included this variable, 11 participants were excluded: 6, EHISS 2014, and 5, EHISS 2020; for responding NS/NC to item Q.87a.2).

Subsequently, two dichotomous variables were created:

*Pain Status* from item Q.45: No; for those who answered "None", or Yes; for those who answered: "Very Mild, Mild, Moderate or Severe, Extreme", to this item).

*Pain Affect* from item Q.46: No, for those who answered "Nothing", or Yes, for those who answered "A little, Moderately, Fairly, or A lot" to this item).

## Statistical analysis

Normal data distribution was checked with the Kolgomorov-Smirnov test (with significance correction of Lilliefors). A descriptive analysis was performed, presenting the continuous variable, age, by median and interquartile range (IQR), and the categorical variables: Sex, BMI, PAF in leisure time, Pain Level, Pain Affect Level, Pain Medication, Pain Status, Pain Affect, Social Class and Civil Status, by their absolute and relative frequencies. We tested the differences between the continuous variable, age, of men and women with the Mann-Whitney U test. The possible relationships between sex and the other categorical variables mentioned above were studied by performing a Chi-Square test. A post-Hoc analysis with z test for independent proportions was performed to evaluate the possible differences between the proportions presented by sex. To interpret size, coefficients were calculated: Phi (2x2 contingency tables) or Cramer's V (contingency tables larger than 2x2); as needed according to the following interpretation: Negligible (0.0–0.10), weak (0.10–0.20), moderate (0.20–0.40), relatively strong (0.40–0.60), strong (0.60–0.80), very strong (0.80–1.00) [26]. We also used chi-squared test, with the post-Hoc tests cited above to evaluate, the associations between PAF in leisure time and three mains pain-related variables: Pain Medication, Pain Status, and Pain Affect. In addition, a correlation study was performed between the PAF in leisure time and the pain-related variables, calculating Spearman's rho, with Bonferroni correction.

Finally, a multiple binomial logistic regression analysis was performed to assess the relation of: pain medication use, pain and pain impairment, each of these being the dependent variable in each one of the models, and the predictor variables entered in each model were: PAF in leisure time, adjusted by Sex, Age, BMI Group, Social Class and Civil Status. A regressive elimination procedure was used to remove all variables that did not influence the dependent variables ($p > 0.05$). Also, the exponential value of B (Exp(B)) that represents the predicted change in odds for a one unit change in the predictor were calculate. For all logistic binary regressions, the assumptions of the absence of influential factors were tested using Cook's distance and the absence of outliers. For the binary logistic regression analysis, the model with the best goodness-of-fit according to the Hosmer-Lemeshow test ($p > 0.05$) was selected.

The significance level for the analyses was established at $p < 0.05$. The analyses were performed with IBM SPSS Statistical v.25, Armonk, NY, USA.

## Results

### Descriptive analysis and relationship between PAF in leisure time and pain-related variables

Data of the variables did not follow normality, $p < 0.001$. Additional file 2 and 3 show the descriptive analysis of the sample from both the EHISS 2014 and EHISS 2020, respectively. They present the median age and the IQR presented by the sample, and by sex. As well as the

absolute and relative frequencies presented by the sample, and by sexes in the study variables: PAF in leisure time, Pain Level, Pain affect Level, Pain medication, Pain status, Pain affect, in addition to other demographic and socio-economic variables such as BMI Group, Social Class and Civil Status. These same tables show the results of the relationships between sex and the variables of interest. Significant relationships (p<0.001) were found in all of them, in the EHISS 2014, as well as in the EHISS 2020. Men presented higher proportions than women did in the highest PAF in leisure time (EHISS 2014. Very active: 15.1% vs 9.7%, p<0.05; EHISS 2020. Very active: 17.5% vs 13.9%, p<0.05) in both surveys. On the other hand, in both surveys' women presented higher proportions of: high levels of pain (EHISS 2014. Extreme: 1.4% vs 0.8%, p<0.05; EHISS 2020. Extreme: 1.2% vs 0.4%, p<0.05), high levels of pain affect (EHISS 2014. A lot: 3.5% vs 2.0, p<0.05; EHISS 2020. A lot: 3.0% vs 1.4%, p<0.05), the use of pain medication (EHIS 2014: 43.7% vs 26.3%, p<0.05; EHISS 2020: 38.0% vs 23.2%, p<0.05), pain status (EHISS 2014: 53.0% vs 36.7%, p<0.05; EHISS 2020: 48.3% vs 33.7%, p<0.05) and pain affect (EHISS 2014: 36.0% vs 22.3%, p<0.05; EHIS 2020: 35.2% vs 22.3%, p<0.05).

Fig 2 shows the prevalence of: pain, pain affect, and use of pain medication, according to the PAF in leisure time of the population of the EHISS 2014 and EHISS 2020. The inactive population in both EHISS 2014 (53.0%) and EHISS 2020 (46.8%) presented higher prevalence of pain, than the active population (37.2% and 33.4%; EHISS 2014 and EHISS 2020, respectively), p<0.05, and then the very active population (34.9% and 33.6%; EHISS 2014 and EHISS 2020, respectively), p<0.05 (Additional file 4). Relationships were found between pain prevalence and PAF in leisure time in the EHISS 2014 ($X^2$ = 340.2, df = 3, p<0.001, V = 0.130) and EHISS 2020 ($X^2$ = 90.9, df = 3, p<0.001, V = 0.105) populations, these dependency relationships, as well as those found in men and women, are shown in additional file 4. The highest prevalence of people who reported having some type of pain affectation was found in the inactive population, both in the EHISS 2014 (39.0%), and in the EHISS 2020 (36.4%). However, the lowest prevalence were found in the active population (20.4% and 20.8%; EHISS 2014 and EHISS 2020, respectively), p<0.05, and in the very active population (18.6% and 20.3%; EHISS 2014 and EHISS 2020, respectively), p<0.05 (Additional file 5). Also, we found dependency relationships between pain affect and PAF in leisure time, as well in EHISS 2014 ($X^2$ = 554.0, df = 3, p<0.001, V = 0.166), like in EHISS 2020 ($X^2$ = 376.4, df = 3, p<0.001, V = 0.140). We detected the same dependency relationships when stratifying the sample by sex, p<0.001 (Additional file 5). The highest prevalence of medication use for pain was also found in the inactive population, in both EHISS 2014 (42.1%), and EHISS 2020 (36.7%), while the lowest were found in the active (28.5% and 22.8%; EHISS 2014 and EHISS 2020, respectively), p<0.05, and very active (29.1% and 24.4%; EHISS 2014 and EHISS 2020, respectively), p<0.05 (Additional file 6). In addition, we found dependency relationships between medication use for pain and PAF in leisure time, as well in EHISS 2014 ($X^2$ = 237.5, df = 3, p<0.001, V = 0.109), like in EHISS 2020 ($X^2$ = 228.8, df = 3, p<0.001, V = 0.109), we detected the same dependency relationships when stratifying the sample by sex, p<0.001 (Additional file 6).

**Correlational analysis.** Table 1 shows the correlations between PAF in leisure time and pain-related variables (Pain, Pain Level, Pain Affect, Pain Affect Level, and Pain Medication) in the EHISS 2014 and EHISS 2020. In all variables studied, negative weak but significant negative relationships were found indicating that Weak but significant negative relationships were found for all the variables studied, indicating that lower PAF in leisure time are associated with the higher prevalence of pain-related variables.

**Multiple binary logistic regression.** The results of the multiple binary logistic regression model are presented in Table 2. Also presented the exponential value of B (Exp(B)) that represents the predicted change in odds of dependent variable and for a one unit change in the predictor. Being active decreases the odds of pain by 0.739 (95% CI 0.668–0.819) and 0.757 (95%

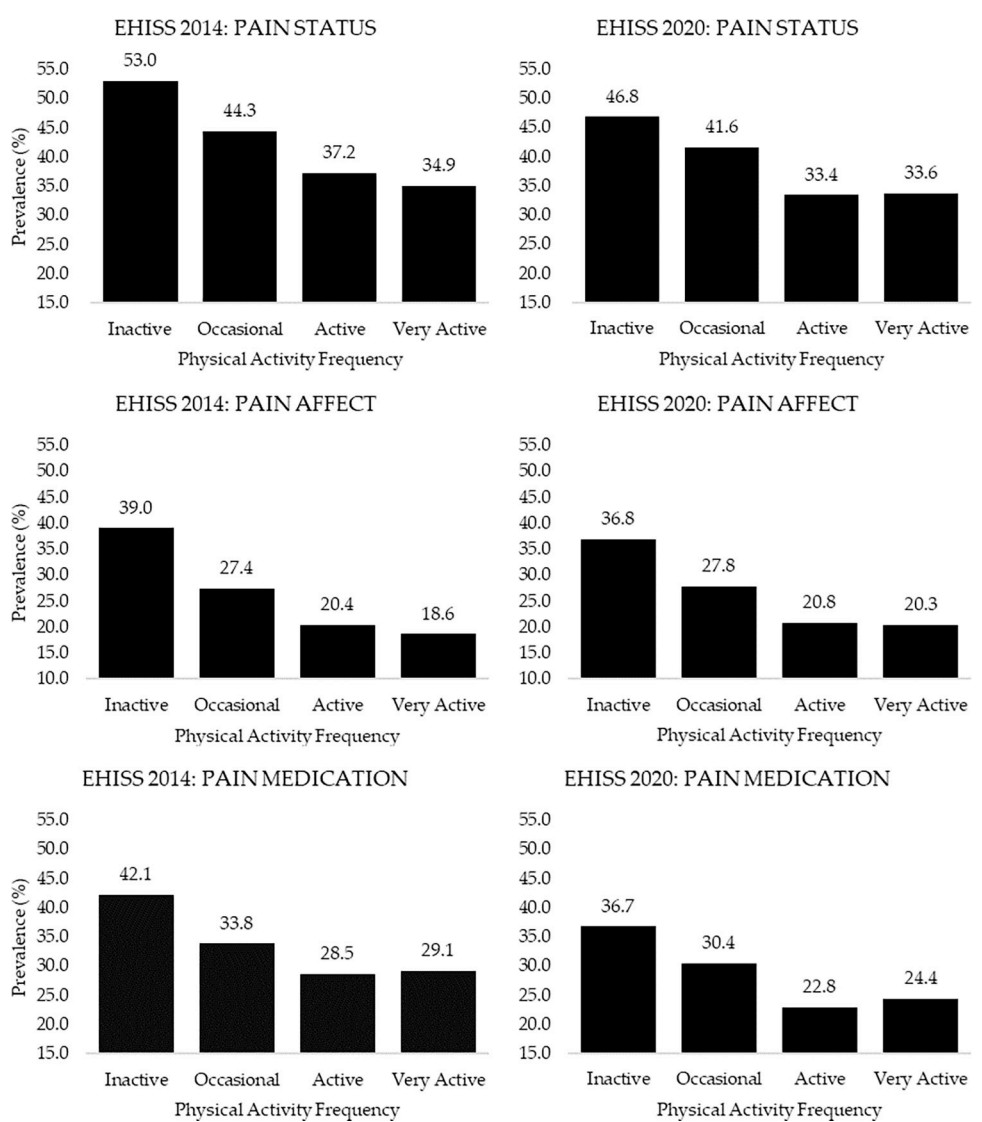

**Fig 2. Prevalence of pain status, pain affect, and pain medication, according to physical activity frequency in Spanish population from European Health Interview Survey of Spain (EHISS) 2014 and 2020.** The bars show the relative frequencies of the answers "Yes" in the three variables.

CI 0.676–0.846) times in relation to inactive people in EHISS 2014 and EHISS 2020, respectively. Being very active decreases the odds of pain by 0.664 (95% CI 0.600–0.736) and 0.776 (95% CI 0.703–0.856) times in EHISS 2014 and EHISS 2020, respectively in presence of the rest the predictors variables. In addition, the multiple binary logistic regression shows that being female increases the odds of pain by 1.973 (95% CI 1.856–2.098) and 1.866 (95% CI 1.751–1.989) times in relation to men in EHISS 2014 and EHISS 2020, Also and being from the lowest socioeconomic class (VI) increases the odds of pain by 1.239 (95% CI 1.098–1.397) and 1.409 (95% CI 1.240–1.601) times in relation to highest socioeconomic class (I) in EHISS 2014 and EHISS 2020, respectively in presence of the rest the predictors variables. For each additional year of age increases the odds of pain by 1.021 (95% CI 1.019–1.023) and 1.023 (95% CI 1.021–1.025) times in EHISS 2014 and EHISS 2020, respectively. Finally, having obesity increases the odds of pain 1.355 (95% CI 1.076–1.708) and 1.210 (95% CI 0.948–1.545)

**Table 1. Relationships between frequency of physical activity in leisure time and prevalence of pain status, pain level, pain affect, pain affect level, and use of pain medication in the Spanish population in European Health Interview Survey of Spain (EHISS) 2014 and 2020.**

**2014**

**Physical Activity Frequency**

| Variables | Total | | Men | | Women | |
|---|---|---|---|---|---|---|
| | rho | p | rho | p | rho | P |
| Pain Status | -0.130 | <0.001 | -0.133 | <0.001 | -0.088 | <0.001 |
| Pain Level | -0.156 | <0.001 | -0.154 | <0.001 | -0.120 | <0.001 |
| Pain Affect | -0.165 | <0.001 | -0.101 | <0.001 | -0.102 | <0.001 |
| Pain Affect Level | -0.177 | <0.001 | -0.171 | <0.001 | -0.134 | <0.001 |
| Pain Medication | -0.107 | <0.001 | -0.037 | <0.001 | -0.067 | <0.001 |

**2020**

**Physical Activity Frequency**

| Variables | Total | | Men | | Women | |
|---|---|---|---|---|---|---|
| | rho | p | Rho | p | rho | P |
| Pain | -0.102 | <0.001 | -0.096 | <0.001 | -0.089 | <0.001 |
| Pain Level | -0.117 | <0.001 | -0.104 | <0.001 | -0.111 | <0.001 |
| Pain Affect | -0.139 | <0.001 | -0.122 | <0.001 | -0.138 | <0.001 |
| Pain Affect Level | -0.151 | <0.001 | -0.128 | <0.001 | -0.155 | <0.001 |
| Pain Medication | -0.107 | <0.001 | -0.076 | <0.001 | -0.113 | <0.001 |

Rho: (Spearman correlation coefficient) with Bonferroni correction factor. The significance level set at p<0.001.

times in relation to underweight in EHISS 2014 and EHISS 2020, respectively also in presence of the rest the predictors variables (Table 2).

Table 2 also show the rest of multiple binary logistic regression model results for pain deterioration, and analgesic use with the EHISS 2014 and EHISS 2020 data.

## Discussion

The main objective of this study was to analyze the relationship between the frequency of physical activity in the leisure time and pain-related variables such as the presence of Pain, Pain Level, Pain Affect, Pain Affect Level, Pain Medication in a sample of the general Spanish population aged between 18 and 79 years both in the general population and by sex.

### Frequency of physical activity in leisure time and pain-related variables

The main results of this study support our initial hypothesis, showing significant relations between PAF in leisure time and pain-related variables in both the general population and by sex of our sample. Specifically, we found that the highest prevalence (between 36.7% and 53%) of Pain, Pain Status, limitations, or impairment in daily activities due to pain and Pain medication were found in those who were physically inactive. In contrast, in our sample those who were active or very active, the prevalence obtained for these variables (Pain, Pain Status, limitations, or impairment in daily activities due to pain and Pain medication) were the lowest (between 18.6% and 37.2%). Furthermore, the same dependence relationships between PAF in leisure time and pain-related variables were found in both men and women.

Similar to our results, several studies in multiple populations have shown a negative relationship between pain scores and physical activity, with more active people having lower pain scores compared to more inactive people [27–29]. These studies have also observed this

**Table 2. Multiple binary logistic regression model for pain, affect pain, and pain medication use.**

| | | EHISS 2014 | | | | EHISS 2020 | | | |
|---|---|---|---|---|---|---|---|---|---|
| | | **Model for Pain** | | | | | | | |
| | | Sig. | Exp(B) | 95% C.I.for EXP(B) | | Sig. | Exp(B) | 95% C.I.for EXP(B) | |
| | | | | Lower | Upper | | | Lower | Upper |
| PAF: Inactive | Occasional | <0.001 | 0.693 | 0.647 | 0.743 | <0.001 | 0.853 | 0.794 | 0.917 |
| | Active | <0.001 | 0.739 | 0.668 | 0.819 | <0.001 | 0.757 | 0.676 | 0.846 |
| | Very active | <0.001 | 0.664 | 0.600 | 0.736 | <0.001 | 0.776 | 0.703 | 0.856 |
| Social Class: I | II | 0.546 | 1.042 | 0.912 | 1.189 | 0.918 | 1.008 | 0.874 | 1.162 |
| | III | 0.123 | 1.090 | 0.977 | 1.217 | 0.062 | 1.116 | 0.995 | 1.252 |
| | IV | 0.014 | 1.157 | 1.029 | 1.300 | 0.029 | 1.148 | 1.014 | 1.300 |
| | V | 0.009 | 1.146 | 1.034 | 1.270 | <0.001 | 1.252 | 1.123 | 1.396 |
| | VI | 0.001 | 1.239 | 1.098 | 1.397 | <0.001 | 1.409 | 1.240 | 1.601 |
| Sex (Men) | Women | <0.001 | 1.973 | 1.856 | 2.098 | <0.001 | 1.866 | 1.751 | 1.989 |
| Age | Age (Years) | <0.001 | 1.021 | 1.019 | 1.023 | <0.001 | 1.023 | 1.021 | 1.025 |
| BMI: Underweight | Normal | 0.278 | 0.885 | 0.709 | 1.104 | 0.042 | 0.783 | 0.619 | 0.991 |
| | Overweight | 0.796 | 1.030 | 0.823 | 1.289 | 0.474 | 0.917 | 0.723 | 1.163 |
| | Obesity | 0.010 | 1.355 | 1.076 | 1.708 | 0.126 | 1.210 | 0.948 | 1.545 |
| | Constant | <0.001 | 0.226 | | | <0.001 | 0.168 | | |
| | | **Model for Pain Affect** | | | | | | | |
| PAF: Inactive | Occasional | <0.001 | 0.585 | 0.544 | 0.630 | <0.001 | 0.697 | 0.645 | 0.753 |
| | Active | <0.001 | 0.597 | 0.531 | 0.671 | <0.001 | 0.609 | 0.536 | 0.692 |
| | Very active | <0.001 | 0.524 | 0.465 | 0.591 | <0.001 | 0.611 | 0.547 | 0.683 |
| Social Class: I | II | 0.654 | 1.036 | 0.888 | 1.209 | 0.576 | 1.048 | 0.889 | 1.236 |
| | III | 0.012 | 1.176 | 1.036 | 1.335 | 0.014 | 1.180 | 1.034 | 1.348 |
| | IV | 0.001 | 1.261 | 1.104 | 1.440 | <0.001 | 1.350 | 1.173 | 1.554 |
| | V | <0.001 | 1.342 | 1.193 | 1.510 | <0.001 | 1.454 | 1.284 | 1.647 |
| | VI | <0.001 | 1.421 | 1.241 | 1.627 | <0.001 | 1.633 | 1.416 | 1.883 |
| Sex (Men) | Women | <0.001 | 1.975 | 1.844 | 2.114 | <0.001 | 1.909 | 1.780 | 2.048 |
| Age | Age (Years) | <0.001 | 1.023 | 1.021 | 1.026 | <0.001 | 1.027 | 1.024 | 1.029 |
| BMI: Underweight | Normal | 0.344 | 1.135 | 0.873 | 1.475 | 0.094 | 0.802 | 0.619 | 1.039 |
| | Overweight | 0.014 | 1.396 | 1.071 | 1.819 | 0.424 | 0.899 | 0.693 | 1.167 |
| | Obesity | <0.001 | 1.822 | 1.391 | 2.387 | 0.146 | 1.219 | 0.934 | 1.592 |
| | Constant | <0.001 | 0.074 | | | <0.001 | 0.076 | | |
| | | **Model for Pain Medication** | | | | | | | |
| PAF: Inactive | Occasional | <0.001 | 0.728 | 0.678 | 0.780 | <0.001 | 0.786 | 0.729 | 0.848 |
| | Active | <0.001 | 0.734 | 0.660 | 0.817 | <0.001 | 0.661 | 0.585 | 0.747 |
| | Very active | <0.001 | 0.755 | 0.679 | 0.839 | <0.001 | 0.734 | 0.660 | 0.815 |
| Social Class: I | II | 0.058 | 0.874 | 0.761 | 1.004 | 0.807 | 0.981 | 0.839 | 1.147 |
| | III | 0.698 | 0.978 | 0.872 | 1.096 | 0.206 | 1.084 | 0.957 | 1.229 |
| | IV | 0.327 | 1.062 | 0.941 | 1.198 | 0.006 | 1.206 | 1.055 | 1.379 |
| | V | 0.514 | 1.036 | 0.931 | 1.153 | <0.001 | 1.322 | 1.175 | 1.487 |
| | VI | 0.065 | 1.124 | 0.993 | 1.274 | <0.001 | 1.433 | 1.250 | 1.641 |
| Sex (Men) | Women | <0.001 | 2.202 | 2.063 | 2.351 | <0.001 | 2.073 | 1.937 | 2.219 |
| Age | Age (Years) | <0.001 | 1.008 | 1.006 | 1.011 | <0.001 | 1.014 | 1.012 | 1.016 |
| BMI: Underweight | Normal | 0.870 | 0.981 | 0.780 | 1.234 | 0.509 | 0.919 | 0.716 | 1.180 |
| | Overweight | 0.093 | 1.221 | 0.967 | 1.540 | 0.400 | 1.115 | 0.866 | 1.435 |
| | Obesity | 0.001 | 1.509 | 1.189 | 1.916 | 0.002 | 1.497 | 1.156 | 1.940 |

*(Continued)*

**Table 2.** (Continued)

| | | EHISS 2014 | | | | EHISS 2020 | | | |
|---|---|---|---|---|---|---|---|---|---|
| | | **Model for Pain** | | | | | | | |
| | | Sig. | Exp(B) | 95% C.I.for EXP(B) | | Sig. | Exp(B) | 95% C.I.for EXP(B) | |
| | | | | Lower | Upper | | | Lower | Upper |
| Social Status: Single | Married | 0.003 | 1.128 | 1.042 | 1.222 | - | - | - | - |
| | Widowed | 0.007 | 1.214 | 1.053 | 1.400 | - | - | - | - |
| | Legally Separated | 0.969 | 0.996 | 0.818 | 1.214 | - | - | - | - |
| | Divorced | 0.005 | 1.235 | 1.064 | 1.432 | - | - | - | - |
| | Constant | <0.001 | 0.222 | | | <0.001 | 0.133 | | |

EHISS (European Health Interview Survey in Spain); Sig (Statistical significance); Exp (Exponential regression); CI (Confidence Interval);—(Not applicable).

relationship in both sexes, in different types of pain and analyzing different states of fitness or intensity of physical activity [27–31]. In our study, only the frequency of physical activity in the leisure time is taken into account, i.e., how sedentary or inactive people are or how active people are in your leisure time. The results show that people who performed physical activity several times a month or a week shows with a lower prevalence of pain and pain-related variables. These results reinforce the previous findings reported in a study which analyzed the Spanish National Health Survey 2017 in 17,777 Spanish residents [20]. In this study, inactive people also had the highest prevalence of pain, were more affected by pain in their daily lives, and took more medication [20]. In addition, also similar results also show the study by Hashem et al. [32] in which physically active people were associated with a lower frequency of back pain than physically inactive people. Also, in a previous study, the main finding was an inverse association between exercise and chronic pain: the more frequent the physical exercise, the lower the risk of chronic pain [33].

Given the high prevalence of people suffering from some type of pain [34], it is important to continue research and to understand in greater depth the relation between these variables and the impact that physical activity could have on pain. It should be noted that, the relation between PA and pain is bidirectionality [35]. In addition to the negative impact of pain on general health, pain can also lead to maladaptive behaviors, such as a reduction in physical activity in people with pain [36, 37]. In this sense, and also in line with our results, several studies report lower levels of physical activity in people with chronic pain [27, 38]. Currently, evidence suggests that physically inactive or sedentary behaviors are in relationship with pain situations, diseases, or pathologies and it is thought that they may contribute to the maintenance in the time of these painful situations [36, 37]. Also, physically inactive behaviors have also been related with poorer general health and poorer mental health, which in turn might be a contributing factor to pain conditions, [39]. Furthermore, our correlational analysis corroborates these previous findings and we also found in our sample weak but significant correlations between PAF in leisure time and the above-mentioned pain-related variables, with higher frequencies of physical activity corresponding to lower levels of these variables. In this line, a previous study obtained the same correlation between the variables of PAF and non-specific low back pain, so that an increase in PAF influences a lower risk of suffering chronic pain [40].

Besides, several studies relationship moderate physical activity with less musculoskeletal pain [31, 39]. Therefore, some authors suggest that physical inactivity may be a risk factor for pain [13, 41]. On the other hand, several authors also report that physical activity may have protective effects against pain and its consequences [42].

## Frequency of physical activity in leisure time, sex, IBM, social class, age

In a first exploratory analysis, our results confirm previous findings in several studies in which a lower frequency of physical activity is observed in women [43, 44]. Among the possible causes or factors, it is discussed that self-efficacy, social support and motivation are factors that influence the lower participation in physical activity among women compared to men [44]. Also, as other studies reported, higher prevalence of pain and pain intensity as well as the use of medication were reported in women, who were also more affected in their daily life due to pain [20, 43].

In addition, to further investigate the factors that influences in the prevalence of pain, pain affect, or medication use in our sample of general Spanish population beyond physical activity, a binary multiple logistic regression analysis was performed given other factors, such as age, sex, BMI, social class and civil status.

The results show that physical inactivity, low social class (VI), female sex, older age and obesity were the factors most likely to increase the odds of pain, pain affect, and analgesic use in both surveys. Our results are partially consistent with those reported by Nils et al. [45]. The authors in their study found that higher levels of physical activity decreased the odds of reporting pain, and being older, obese, female, and having low-income predisposed participants to report musculoskeletal pain 10 years later [45].

Although in our study we only addressed the relationship between recreational PAF, pain and pain-related variables in a Spanish population sample and therefore did not establish causality between these variables, it should be noted that there is growing evidence to suggest that physical activity has a positive impact not only on cardiovascular health but also improves mood, decreases stress and has multiple benefits on mental and general health [9, 46]. It is therefore currently an effective and indispensable therapeutic approach in pain patients as it not only has positive effects on pain modulation but also has a positive impact on the quality of life of these patients, increasing self-efficacy, self-confidence, and decreasing maladaptive behaviors or beliefs related to pain and movement [28, 31, 43].

The mechanisms underlying the impact of physical activity on pain continue to be debated. Among others, it has been suggested that physical activity modulates pain by activating central pain inhibitory pathways, decreasing cortisol, reducing glial cell activation, reducing inflammatory cytokines, and increasing anti-inflammatory cytokines [42, 47].

Regarding sex, in line with our results, several studies have reported that being female increases the odds of pain [39, 45]. Although the underlying mechanisms are currently unclear, hormonal, biological and psychosocial differences, such as greater catastrophic thinking about pain and lower self-efficacy, which have been associated with greater pain, are discussed as possible factors [48].

In respect of the relationship between obesity and pain, at present there is still no clear certainty and a bi-directionality between them cannot be ruled out [45]. However, several hypotheses are under discussion: on the one hand, excess weight may predispose to the development of osteoarthritis, mainly in the lower back, knees, or hips [45, 49]. On the other hand, obese people have high levels of inflammatory markers such as interleukin 6 or C-reactive protein, which could be generating a hyperalgesic state [45, 49]. Therefore, some authors suggest that reducing obesity could have a positive effect on patients' pain and that this in turn could help people to increase their physical activity [45].

Finally, social classes based on occupational occupation classified as VI, i.e. unskilled workers, also had higher odds for pain status, pain affect and pain medication. It might be thought that people in this social class have lower incomes than those in social class I (Directors and

managers of establishments with 10 or more employees and professionals traditionally associated with university degrees). In this sense, low income is known to be an economic stressor, which in turn is also a risk factor for developing depressive symptoms and poor general health, which may secondarily influence pain [50]. For example, the study of Müller et al. [51] also reported a higher likelihood of headaches in those with low socio-economic status. This study also found an interaction between obesity and socioeconomic status [51], in the same line that our results.

The results of this study could have important public health implications, helping to promote preventive public health programs in the general Spanish population with pain, especially in people of more age, women, those with a higher BMI and lower socioeconomic class. Based on the results obtained, it seems important to promote regular physical activity during periods of work or leisure inactivity. Likewise, it seems necessary to promote healthy health styles that help to maintain a healthy BMI in this population.

A strength of this study lies in the large sample analyzed, as well as in the type of analysis performed. In our logistic regression analysis, we have taken into account a large number of possible effect-modifying variables such as BMI, social class, sex and age, which several studies have found to be related to low back or neck pain [52, 53]. However, a related limitation is that it has not been possible to control the results for other variables that influence pain and its related variables such as smoking or psychosocial factors [52, 53].

## Limitations

Our study has the inherent weaknesses of cross-sectional designs, which do not allow us to establish causal relationships between factors. Also, several limitations are found in the quantification of physical activity. In relation to this point, the authors consider it appropriate to highlight that although some authors have argued that accelerometry may offer a better assessment of physical activity, than subjective self-report among some issues as it avoids problems of recall bias [54], there are certain notable limitations to objective assessments of physical activity. Between other, dependence on the availability of these devices which in turn implies a smaller sample capacity to be analyzed or dependence on the location of the accelerometer (e.g. wrist vs. hip) [55]. In this regard, it has been reported that a location on the hip would be more advisable as well as placing more than one on different parts of the body [55]. On the other hand, although it seems that respondents tend to overestimate their physical activity, validation studies have shown that the questionnaires are able to rank order individuals according to activity level, so that, in other words, within a sample, they are able to discern who is less or more physically active [56, 57], as well as being cheap, easy instruments that allow large samples of people to be assessed [57]. For all these reasons, we do not consider the use of self-reported tools in our study to have been a limitation.

One aspect to highlight is that the questionnaire question only refers to the amount of physical activity carried out in leisure time. It would be relevant to be able to identify how much time each person spends 24 hours a day on physical activity, sedentary behaviors and sleeping, as each person's free time may vary. This would lead to a more objective analysis of the relationship between pain and each of the behaviors related to daily lifestyle.

Finally, it should be borne in mind that on 17 March 2020 confinement was established in Spain. From that moment on, the interviews went from being face-to-face to telephone interviews, which could have affected the answers provided in some way.

Taking into account all of the above, the authors consider that future studies should be aimed at designing preventive programs based on physical activity to evaluate the impact on pain levels and side effects and on quality of life of Spanish population.

## Conclusions

Physical inactivity was related to a higher prevalence of: pain, higher levels of pain, higher pain involvement and higher analgesic use. In addition, lower social class, being female, older age, and obesity were factors for higher odds of pain, pain involvement, and analgesic use in both surveys.

## Supporting information

**S1 File. Description of social classes based on occupational occupation.**
(DOCX)

**S2 File. Descriptive analysis of the Spanish population of the European Health Survey in Spain 2014.**
(DOCX)

**S3 File. Descriptive analysis of the Spanish population of the European Health Survey in Spain 2020.**
(DOCX)

**S4 File. Relationship between level of physical activity and prevalence of pain in the Spanish population.**
(DOCX)

**S5 File. Relationship between level of physical activity and prevalence of pain affect in Spanish population.**
(DOCX)

**S6 File. Relationship between level of physical activity and prevalence of pain medication use in the Spanish population.**
(DOCX)

**S7 File. Databases.**
(ZIP)

## Author Contributions

**Formal analysis:** Ángel Denche-Zamorano, José Carmelo Adsuar-Sala.

**Investigation:** Ángel Denche-Zamorano, José Carmelo Adsuar-Sala.

**Methodology:** Ángel Denche-Zamorano.

**Resources:** Ángel Denche-Zamorano, Antonio Castillo-Paredes, Diana Salas-Gómez.

**Supervision:** Antonio Castillo-Paredes, Diana Salas-Gómez.

**Visualization:** Antonio Castillo-Paredes, José Carmelo Adsuar-Sala.

**Writing – original draft:** Raquel Pastor-Cisneros, Diana Salas-Gómez.

**Writing – review & editing:** Antonio Castillo-Paredes, Diana Salas-Gómez.

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
