## [Decision Letter · Decision Letter 0]

25 Jan 2024

PONE-D-23-36610Relationships between Physical Activity Frequency in leisure time and: pain, pain impairment and pain medication use in the Spanish adult population. A cross-sectional studyPLOS ONE

Dear Dr. Castillo-Paredes,

Thank you for submitting your manuscript to PLOS ONE. After careful consideration, we feel that it has merit but does not fully meet PLOS ONE’s publication criteria as it currently stands. Therefore, we invite you to submit a revised version of the manuscript that addresses the points raised during the review process. **The points raised by both the reviewers is very important to improve the quality of the paper, hence these comments has to be incorporated in the revised manuscript.  **

We look forward to receiving your revised manuscript.

Kind regards,

Strong P Marbaniang

Academic Editor

PLOS ONE

Journal Requirements:

The APC was funded by the Open Access Program of Universidad de Las Américas

The author A.D.-Z. (FPU20/04201) was supported by a grant from the Spanish Ministry of Education, Culture and Sport. Grants FPU20/04201 funded by MCIN/AEI/10.13039/501100011033 and, as appropriate, by “European Social Found Investing in your future” or by “European Union NextGenerationEU/PRTR”. The author R.P.-C. was supported with a grant by the Valhondo Calaff Foundation (Caceres, Spain)We also thank the Universidad de Las Américas for their support of the Open Access initiative.

The APC was funded by the Open Access Program of Universidad de Las Américas

6. Please include your tables as part of your main manuscript and remove the individual files. Please note that supplementary tables (should remain/ be uploaded) as separate "supporting information" files.

Reviewers' comments:

Reviewer's Responses to Questions

**Comments to the Author**

1. Is the manuscript technically sound, and do the data support the conclusions?

Reviewer #1: Yes

Reviewer #2: Yes

2. Has the statistical analysis been performed appropriately and rigorously? 

Reviewer #1: No

Reviewer #2: Yes

3. Have the authors made all data underlying the findings in their manuscript fully available?

Reviewer #1: No

Reviewer #2: Yes

4. Is the manuscript presented in an intelligible fashion and written in standard English?

Reviewer #1: No

Reviewer #2: Yes

5. Review Comments to the Author

Reviewer #1: Theoretical Level:

1. Clarity and Depth in Theoretical Framework:

o The article presents a solid theoretical framework but lacks a more explicit connection between the theories presented and their application in the study.

o It is suggested that the theoretical discussion be expanded to highlight how existing theories relate directly to the research in question. This will help strengthen the conceptual basis of the study and give coherence to the analysis.

Methodological Level:

2. Details and Clarity of Methodology:

o The methodological description is fairly concise and could be more detailed to allow for accurate replication of the study by other researchers.

o It is recommended that the methodological section be expanded to include more information about the study design, sample selection, instruments used and specific procedures. This will increase the reliability and validity of the results.

Bibliographical references:

3. Updating and Extending References:

o Bibliographic references are adequate but could benefit from more diversity and updating, especially in relation to the most recent developments in the field.

o Seek out and add new and recent research sources that support and enrich the arguments presented. This will strengthen the credibility of the article by demonstrating a thorough understanding of the current state of the field.

Reviewer #2: The manuscript examines the how physical activity can influence pain. I find the paper well-written, and it has the potential to contribute to the existing literature. However, I have some comments I would like the authors to consider.

INTRODUCTION:

The present study is conducted in the Spanish population. In line 64 to 66, authors relate high treatment costs and the access to treatment with social classes, but in Spain there is a public health system that covers the major part of the costs and is accessible for the whole population.

MATERIAL AND METHODS:

Why do the authors conduct the study with data from interviews of two different years?

In line 109 authors refers to the methodology of the surveys, could you add the reference where that methodology can be find?

The exclusion and inclusion criteria are explained in the participants section, but in the flowchart depicted in Figure 1 there are more exclusion criteria that are not explained in the participants section. In my opinion, all the exclusion criteria should be explained in that section.

In line 132-133 it is written: Overweight (BMI between 25 and <30), and Obesity (BMI >30). In which group are included those with BMI=30?

It should be explained that the variable pain medication is a combination of two items and that this variable is not an item of the Interview. It is confusing that the authors just consider “created variables” Pain Status and Pain affect, while pain medication is also created by the authors.

In my opinion, line 212 should be written as: “the significance level for the analyses was established at 0.05” as the significance level is a number, not an interval.

RESULTS:

A lot of information of the results is shown in additional files, some relevant results could be move to the main manuscript.

Could you please specify the p value? Writing p<0.05 does not bring much more information than that the result is significant.

Authors wrote the OR value, but they do not interpret the meaning of that value. That information is important and it should be explained in the results. Furthermore, when explaining the results, authors wrote: “being female increases the odds of pain (OR:1.973; 95 IC 1.856-2.098) and (OR: 1.866; 95 IC 1.751-1.989) times” What does that “times” mean? This is one example, but it is repeated in several sentences.

TABLES AND FIGURES:

Figures are clear and help to read and follow the information given in the manuscript. In Figure 1 y-axes represent different numbers, some of them represent from 0 to 55 and others from 0 to 45.

In Table 1 It is written: “The significance level set at p<0.001” but in the Material and Methods section it says that the significance level is set at 0.05.

There is a mistake in Additional file 6, It is written “Addiotional”

In some Additional files differences that are significant are highlighted with abc, Why do you use different letters? Is there any difference between cells highlighted with a, with b or with c? Furthermore, in other files the significant cells are highlighted with *. I would suggest to the authors that they unify criteria.

DISCUSSION:

The following sentence is unclear: “In contrast, in our sample those who were 288 active or very active, the prevalence obtained for these variables were the lowest” Which variables do the authors refer to?

Please clarify the following sentence (line 291): “several studies in multiple populations have shown an relations between 292 pain scores and physical activity”

Finally, in sentence 296 authors refer to “people who were active several times a month or a week”. People who are considered active should do exercise continuously, someone who do exercise one week and not the next, or one month and not the next month should not be considered active.

Finally, In the limitations authors explain the possible bias due to the outbreak in Spain and its possible impact in the interview, but there is a Spanish National Health Interview in 2017. Did the authors consider used that interview? Furthermore, I do not totally understand why authors used interviews from two years if the study could have been conducted with interviews from just one year as they do not make any comparison between results in the different years.

6. PLOS authors have the option to publish the peer review history of their article (what does this mean?). If published, this will include your full peer review and any attached files.

Reviewer #1: No

Reviewer #2: No

---

## [Author Response · Author response to Decision Letter 0]

3 Mar 2024

dear editor and reviewers, 

Documents with responses to your comments are attached. 

Kind regards.

---

## [Editor Report · Decision Letter 1]

24 Mar 2024

PONE-D-23-36610R1Relationships between Physical Activity Frequency in leisure time and: pain, pain impairment and pain medication use in the Spanish adult population. A cross-sectional studyPLOS ONE

Dear Dr. Castillo-Paredes,

Thank you for submitting your manuscript to PLOS ONE. After careful consideration, we feel that it has merit but does not fully meet PLOS ONE’s publication criteria as it currently stands. Therefore, we invite you to submit a revised version of the manuscript that addresses the points raised during the review process.

We look forward to receiving your revised manuscript.

Kind regards,

Strong P Marbaniang

Academic Editor

PLOS ONE

Journal Requirements:

**Additional Editor Comments:**

The authors has to take importance on the theoretical concept of this paper needs to elaborate further as suggested by the reviewer. Also, the reviewer has comment on the validity of the methodology used

Incorporate all the comments and suggestion by the two reviewers

Reviewers' comments:

Reviewer 1:

Theoretical Level:

1. Clarity and Depth in Theoretical Framework:

o The article presents a solid theoretical framework but lacks a more explicit connection between the theories presented and their application in the study.

o It is suggested that the theoretical discussion be expanded to highlight how existing theories relate directly to the research in question. This will help strengthen the conceptual basis of the study and give coherence to the analysis.

Methodological Level:

2. Details and Clarity of Methodology:

o The methodological description is fairly concise and could be more detailed to allow for accurate replication of the study by other researchers.

o It is recommended that the methodological section be expanded to include more information about the study design, sample selection, instruments used and specific procedures. This will increase the reliability and validity of the results.

Bibliographical references:

3. Updating and Extending References:

o Bibliographic references are adequate but could benefit from more diversity and updating, especially in relation to the most recent developments in the field.

o Seek out and add new and recent research sources that support and enrich the arguments presented. This will strengthen the credibility of the article by demonstrating a thorough understanding of the current state of the field.

Reviewer 2:

The manuscript examines the how physical activity can influence pain. I find the paper well-written, and it has the potential to contribute to the existing literature. However, I have some comments I would like the authors to consider.

INTRODUCTION:

The present study is conducted in the Spanish population. In line 64 to 66, authors relate high treatment costs and the access to treatment with social classes, but in Spain there is a public health system that covers the major part of the costs and is accessible for the whole population.

MATERIAL AND METHODS:

Why do the authors conduct the study with data from interviews of two different years?

In line 109 authors refers to the methodology of the surveys, could you add the reference where that methodology can be find?

The exclusion and inclusion criteria are explained in the participants section, but in the flowchart depicted in Figure 1 there are more exclusion criteria that are not explained in the participants section. In my opinion, all the exclusion criteria should be explained in that section.

In line 132-133 it is written: Overweight (BMI between 25 and <30), and Obesity (BMI >30). In which group are included those with BMI=30?

It should be explained that the variable pain medication is a combination of two items and that this variable is not an item of the Interview. It is confusing that the authors just consider “created variables” Pain Status and Pain affect, while pain medication is also created by the authors.

In my opinion, line 212 should be written as: “the significance level for the analyses was established at 0.05” as the significance level is a number, not an interval.

RESULTS:

A lot of information of the results is shown in additional files, some relevant results could be move to the main manuscript.

Could you please specify the p value? Writing p<0.05 does not bring much more information than that the result is significant.

Authors wrote the OR value, but they do not interpret the meaning of that value. That information is important and it should be explained in the results. Furthermore, when explaining the results, authors wrote: “being female increases the odds of pain (OR:1.973; 95 IC 1.856-2.098) and (OR: 1.866; 95 IC 1.751-1.989) times” What does that “times” mean? This is one example, but it is repeated in several sentences.

TABLES AND FIGURES:

Figures are clear and help to read and follow the information given in the manuscript. In Figure 1 y-axes represent different numbers, some of them represent from 0 to 55 and others from 0 to 45.

In Table 1 It is written: “The significance level set at p<0.001” but in the Material and Methods section it says that the significance level is set at 0.05.

There is a mistake in Additional file 6, It is written “Addiotional”

In some Additional files differences that are significant are highlighted with abc, Why do you use different letters? Is there any difference between cells highlighted with a, with b or with c? Furthermore, in other files the significant cells are highlighted with *. I would suggest to the authors that they unify criteria.

DISCUSSION:

The following sentence is unclear: “In contrast, in our sample those who were 288 active or very active, the prevalence obtained for these variables were the lowest” Which variables do the authors refer to?

Please clarify the following sentence (line 291): “several studies in multiple populations have shown an relations between 292 pain scores and physical activity”

Finally, in sentence 296 authors refer to “people who were active several times a month or a week”. People who are considered active should do exercise continuously, someone who do exercise one week and not the next, or one month and not the next month should not be considered active.

Finally, In the limitations authors explain the possible bias due to the outbreak in Spain and its possible impact in the interview, but there is a Spanish National Health Interview in 2017. Did the authors consider used that interview? Furthermore, I do not totally understand why authors used interviews from two years if the study could have been conducted with interviews from just one year as they do not make any comparison between results in the different years.

---

## [Author Response · Author response to Decision Letter 1]

4 Jun 2024

EDITOR

Dear Editor,

Thank you for considering that our article has the potential to be published in your journal. We appreciate the opportunity you have given us to improve our manuscript. We also thank the reviewers for all the comments for improvement.

Attached are the responses to the reviewers.

Best Regards

REVIEWER 1

Dear Reviewer,

Thank you for your review of our manuscript. We have carefully considered your comments and believe that the quality of the paper has improved after incorporating your suggestion. Below is our response to your suggestion:

Theoretical Level:

1. Clarity and Depth in Theoretical Framework:

o The article presents a solid theoretical framework but lacks a more explicit connection between the theories presented and their application in the study.

Author´s response: 

Thank you for your comment. The introduction now includes a paragraph that explicitly connects the scientific evidence provided with the main objective of the study and its application.

o It is suggested that the theoretical discussion be expanded to highlight how existing theories relate directly to the research in question. This will help strengthen the conceptual basis of the study and give coherence to the analysis.

Author´s response: 

Thank you very much for your contribution. The discussion section has been expanded, clearly improving the quality of the research as well as reinforcing the coherence of the explored relationship.

Methodological Level:

2. Details and Clarity of Methodology:

o The methodological description is fairly concise and could be more detailed to allow for accurate replication of the study by other researchers.

It is recommended that the methodological section be expanded to include more information about the study design, sample selection, instruments used and specific procedures. This will increase the reliability and validity of the results.

Author´s response: 

Thank you for your comment. 

Although we had previously included them in the data accessibility section. After your comment we have included in the methodology the links to the databases of our study. From them you will be able to download the database in different formats and you will be able to follow step by step the procedures explained in our manuscript and that we performed in our research.

References 22 and 23 included the survey methodologies, explaining in detail the sample calculation, the sampling system, how missing data were treated and other detailed information about the procedures performed. We included the information necessary for our study to be replicated, but we did not go into the survey methodology in detail so as not to commit plagiarism and not to include information that has already been detailed in the bibliography. If you think we should include any specific information, we would like you to let us know so that we can improve the comprehensibility of our manuscript.

The data are public and open access, which makes it easier for our research to be replicated by any researcher. However, we have made minor changes to this section to improve understanding.

Bibliographical references:

3. Updating and Extending References:

o Bibliographic references are adequate but could benefit from more diversity and updating, especially in relation to the most recent developments in the field.

Author´s response: 

We appreciate your comment. The bibliographical references are adequate and have been fully updated as well as related to more recent developments and research in this field of knowledge.

o Seek out and add new and recent research sources that support and enrich the arguments presented. This will strengthen the credibility of the article by demonstrating a thorough understanding of the current state of the field.

Author´s response: Thank you for your comment. Following your suggestion, new research has been added to enrich the arguments presented in the article.

REVIEWER 2:

Dear Reviewer,

Thank you for your review of our manuscript. We have carefully considered your comments and believe that the quality of the paper has improved after incorporating your suggestion. Below is our response to your suggestion:

INTRODUCTION: 

The present study is conducted in the Spanish population. In line 64 to 66, authors relate high treatment costs and the access to treatment with social classes, but in Spain there is a public health system that covers the major part of the costs and is accessible for the whole population. 

Author´s response: 

Thank you for your comment. The authors have decided to remove the sentence as it does not correspond to the context of the Spanish population.

MATERIAL AND METHODS: 

Why do the authors conduct the study with data from interviews of two different years? 

Author´s response:

The EHISS is carried out periodically to know the health status of the Spanish population. The last two surveys carried out were the EHISS 2014 and the EHISS 2020. We believe that including data from both provides greater strength to the results of our study. This cross-sectional study demonstrates at two different times and with a larger sample how PA was related to pain in the Spanish population.

In line 109 authors refers to the methodology of the surveys, could you add the reference where that methodology can be find?

Author´s response: 

Amended

The exclusion and inclusion criteria are explained in the participants section, but in the flowchart depicted in Figure 1 there are more exclusion criteria that are not explained in the participants section. In my opinion, all the exclusion criteria should be explained in that section. 

Author´s response:

Thank you for your comment. We believe that we had not explained this procedure well. In the "Participants" section we explain the general inclusion and exclusion criteria. On the other hand, in the "Variables" section we explain the exclusions made for the analyzes of specific variables. The flowchart contains both procedures.

Participants who met the inclusion criteria were included in the study. However, if data were not presented on any of the study variables, they were only excluded from the analysis that included said variable.

When citing figure 1 we have added a clarification about it thanks to your contribution.

In line 132-133 it is written: Overweight (BMI between 25 and <30), and Obesity (BMI >30). In which group are included those with BMI=30? 

Author´s response: 

Amended.

It should be explained that the variable pain medication is a combination of two items and that this variable is not an item of the Interview. It is confusing that the authors just consider “created variables” Pain Status and Pain affect, while pain medication is also created by the authors. 

Author´s response: 

We appreciate your comment. A sub-section called created variables has been created and it has been indicated that the variable pain medication corresponds to the combination of two items.

In my opinion, line 212 should be written as: “the significance level for the analyses was established at 0.05” as the significance level is a number, not an interval. 

Author´s response: 

Amended.

RESULTS: 

A lot of information of the results is shown in additional files, some relevant results could be move to the main manuscript.

Author´s response:

Thank you for your comment. We agree with you. Our research includes a lot of important data in the additional material. The additional material is part of the research and can be accessed by readers. We would like you to let us know which table you would like us to include within the manuscript and not as additional material.

Could you please specify the p value? Writing p<0.05 does not bring much more information than that the result is significant. 

Author´s response:

Thank you for your comment. Amended.

Authors wrote the OR value, but they do not interpret the meaning of that value. That information is important and it should be explained in the results. Furthermore, when explaining the results, authors wrote: “being female increases the odds of pain (OR:1.973; 95 IC 1.856-2.098) and (OR: 1.866; 95 IC 1.751-1.989) times” What does that “times” mean? This is one example, but it is repeated in several sentences. 

Author´s response:

Thank you for your comment. That paragraph has been rewritten so that it can be better understood.

TABLES AND FIGURES: 

Figures are clear and help to read and follow the information given in the manuscript. In Figure 1 y-axes represent different numbers, some of them represent from 0 to 55 and others from 0 to 45. 

Author´s response:

Amended.

In Table 1 It is written: “The significance level set at p<0.001” but in the Material and Methods section it says that the significance level is set at 0.05. 

Author´s response:

That was the p value with the Bomferroni correction factor.

There is a mistake in Additional file 6, It is written “Addiotional”

Author´s response:

Amended.

In some Additional files differences that are significant are highlighted with abc, Why do you use different letters? Is there any difference between cells highlighted with a, with b or with c? Furthermore, in other files the significant cells are highlighted with *. I would suggest to the authors that they unify criteria.

Author´s response:

You refer to the results of the post hoc pairwise z test for independent proportions. It is not possible to unify criteria.

* was used when comparisons were made between two groups: Men and Women. If differences are found between the proportions of men and women, it is indicated with an *.

When comparisons were made between PAF groups, “abc” was used. Different letters indicated significant differences between PAF groups. A proportion "a" differs significantly from a proportion "b" and both differ from a proportion "c". On the other hand, if "a", "ab", "b" and "c" appear, the proportion "a" differs significantly from the proportion "b" and "c", and the "b" differs from the "c ". The proportion "bc" differs significantly from the proportion "a", but it does not differ significantly from the proportions "b" and "c".

For example:

DISCUSSION:

The following sentence is unclear: “In contrast, in our sample those who were 288 active or very active, the prevalence obtained for these variables were the lowest” Which variables do the authors refer to?

Author´s response: 

We appreciate your feedback. The variables have been re-specified in this sentence to avoid confusion.

Please clarify the following sentence (line 291): “several studies in multiple populations have shown an relations between 292 pain scores and physical activity”

Author´s response: 

Following your suggestion, the sentence has been rephrased to clarify the relationship between variables.

Finally, in sentence 296 authors refer to “people who were active several times a month or a week”. People who are considered active should do exercise continuously, someone who do exercise one week and not the next, or one month and not the next month should not be considered active.

Author´s response:

Amended. “people who performed physical activity several times a month or a week”.

Finally, In the limitations authors explain the possible bias due to the outbreak in Spain and its possible impact in the interview, but there is a Spanish National Health Interview in 2017. Did the authors consider used that interview? Furthermore, I do not totally understand why authors used interviews from two years if the study could have been conducted with interviews from just one year as they do not make any comparison between results in the different years. 

Author´s response:

The National Health Survey (ENS) and the European Health Survey (EHS) are different surveys. The ENS is a survey carried out at the national level promoted by the Spanish Ministry of Health. The EHS is a survey promoted by the European Statistical Office (Eurostat). We use data from the last two EHS carried out in Spain.

Our objective was not to compare between years, but to analyze the relationships existing at both times. We believe that testing the same hypotheses at two different points in time and with a larger population provides greater strength to our meetings, which is why we use data from the two surveys. We do not include the ENS data as it is a different survey with a different methodology.

---

## [Decision Letter · Decision Letter 2]

12 Aug 2024

PONE-D-23-36610R2Relationships between Physical Activity Frequency in leisure time and: pain, pain impairment and pain medication use in the Spanish adult population. A cross-sectional studyPLOS ONE

Dear Dr. Castillo-Paredes,

Thank you for submitting your manuscript to PLOS ONE. After careful consideration, we feel that it has merit but does not fully meet PLOS ONE’s publication criteria as it currently stands. Therefore, we invite you to submit a revised version of the manuscript that addresses the points raised during the review process.

We look forward to receiving your revised manuscript.

Kind regards,

André Pontes-Silva

Academic Editor

PLOS ONE

**Additional Editor Comments:**

First of all, thank you for submitting your manuscript to Plos One. I would like to congratulate you on your research. I was asked to make the final decision on this article and I noticed that it has already gone through previous revisions. The study has interesting analyses, so please provide the database with the data in unanalyzed values (i.e. absolute values) as a supplementary file (Excel and English).

Reviewers' comments:

Reviewer's Responses to Questions

**Comments to the Author**

1. If the authors have adequately addressed your comments raised in a previous round of review and you feel that this manuscript is now acceptable for publication, you may indicate that here to bypass the “Comments to the Author” section, enter your conflict of interest statement in the “Confidential to Editor” section, and submit your "Accept" recommendation.

Reviewer #2: All comments have been addressed

2. Is the manuscript technically sound, and do the data support the conclusions?

Reviewer #2: Yes

3. Has the statistical analysis been performed appropriately and rigorously? 

Reviewer #2: Yes

4. Have the authors made all data underlying the findings in their manuscript fully available?

Reviewer #2: No

5. Is the manuscript presented in an intelligible fashion and written in standard English?

Reviewer #2: Yes

6. Review Comments to the Author

Reviewer #2: (No Response)

7. PLOS authors have the option to publish the peer review history of their article (what does this mean?). If published, this will include your full peer review and any attached files.

Reviewer #2: No

---

## [Author Response · Author response to Decision Letter 2]

12 Aug 2024

Dear Editor, 

Thank you for your kind consideration of our research and thank you very much for your promptness. 

As requested, we have included the two raw databases. We have provided the databases in .sav format (SPSS statistical data document), these are the two raw databases with the variables in English. 

In addition, in this new round of revision we have addressed the comments of Reviewer 2, modifying a small error in the wording of the results of our regression models. The reviewer also recommended that we include some of the additional tables in the manuscript, although he did not recommend any specific tables. 

After considering the structure of the article, we decided to unify additional tables 7 and 8. These tables showed the results of multiple logistic regression analyses on pain, affected by pain and the use of pain medication.

As these tables are extensive, the authors have simplified and merged them into the current table 2 in the main manuscript.

We have decided on these tables as we believe they provide relevant data that complement the wording of the results and help readers to understand the results.

We hope we have solved all your doubts about our manuscript. 

Best regards

Dear Reviewer 2,

Thank you for your comment and for helping us to improve our manuscript, we have made further modifications following your recommendations.

Thank you to the authors for amended my previous comments and recommendations and congratulations for your work. 

There is just one comment that, from my point of view, that it is not completely solved. It is the one related to the OR and the word “times” when authors describe the results: It does not make sense to put the OR and 95CI between brackets and the word times out of brackets, when this word refers to that information. I think authors should dropped the word times from there. 

Author’s Response:

Thank you for your comment. There was indeed an error in the wording of the results of the regression models and these have been amended.

Regarding the table that can be moved from the additional files to the manuscript, I think authors have to choose the table with more relevant and compelling information for the study.

Author’s Response:

Regarding the tables included in the supplementary material. 

After considering the structure of the article, we decided to unify additional tables 7 and 8. These tables showed the results of multiple logistic regression analyses on pain, affected by pain and the use of pain medication.

As these tables are extensive, the authors have simplified and merged them into the current table 2 in the main manuscript.

We have decided on these tables as we believe they provide relevant data that complement the wording of the results and help readers to understand the results.

The rest of the tables continue in the additional files but in the main manuscript are the figures showing the data from these tables.

The reader who wants to go deeper into them will be able to access the additional material and find what he/she needs. The additional material is part of the research and is accessible to all, we believe that this does not compromise the understanding of the research.

Best Regards

---

## [Editor Report · Decision Letter 3]

19 Aug 2024

PONE-D-23-36610R3Relationships between Physical Activity Frequency in leisure time and: pain, pain impairment and pain medication use in the Spanish adult population. A cross-sectional studyPLOS ONE

Dear Dr. Castillo-Paredes,

Thank you for submitting your manuscript to PLOS ONE. After careful consideration, we feel that it has merit but does not fully meet PLOS ONE’s publication criteria as it currently stands. Therefore, we invite you to submit a revised version of the manuscript that addresses the points raised during the review process.

Dear Dr. Castillo-Paredes, thank you for the corrections and clarifications regarding your article. After careful review, I would like to ask you to kindly make the final adjustments.Your title is long and not very intuitive. I suggest you change it to “Frequency of physical activity during leisure time and its effect on pain variables and medication use: A Spanish Cross-Sectional Study”. Think about whether this makes sense for your study (it is just a suggestion).Organize the abstract into topics (Background [research question]; Objective; Methods; Results; Conclusion).The objective stated in your abstract is different from the objective stated in your introduction. Please correct this. The sentence describing the objective must be exactly the same in both.In the methods section, include a topic called "Sample Size" and explain in detail the sampling strategies used in this study.In the conclusion, objectively describe your final considerations in a single paragraph. Avoid phrases like “Based on the results of this study” or “In our sample the results of this study”. The conclusion described in the abstract and the conclusion described after the discussion must be exactly the same.

We look forward to receiving your revised manuscript.

Kind regards,

André Pontes-Silva

Academic Editor

PLOS ONE
---

## [Author Response · Author response to Decision Letter 3]

2 Sep 2024

We thank the reviewers for all their work, comments and recommendations to improve the presentation of our research.

---

## [Editor Report · Decision Letter 4]

5 Sep 2024

Frequency of Physical Activity During Leisure Time and Variables Related to Pain and Pain Medication Use in Spanish Adults: A Cross-sectional Study

PONE-D-23-36610R4

Dear Dr. Castillo-Paredes,

We’re pleased to inform you that your manuscript has been judged scientifically suitable for publication and will be formally accepted for publication once it meets all outstanding technical requirements.

Kind regards,

André Pontes-Silva

Academic Editor

PLOS ONE

---

## [Editor Report · Acceptance letter]

10 Oct 2024

PONE-D-23-36610R4 

PLOS ONE

Dear Dr. Castillo-Paredes, 

I'm pleased to inform you that your manuscript has been deemed suitable for publication in PLOS ONE. Congratulations! Your manuscript is now being handed over to our production team.

Kind regards, 

on behalf of

Professor André Pontes-Silva 

Academic Editor

PLOS ONE